# Relative influence of intellectual disabilities and autism on mental and general health in Scotland: a cross-sectional study of a whole country of 5.3 million children and adults

Deborah Kinnear, Ewelina Rydzewska, Kirsty Dunn,
Laura Anne Hughes-McCormack, Craig Melville, Angela Henderson,
Sally-Ann Cooper

Institute of Health and Wellbeing, University of Glasgow, Glasgow, UK

**Correspondence to**
Dr Deborah Kinnear;
deborah.kinnear@glasgow.ac.uk

## ABSTRACT

**Objectives** To determine the relative extent that autism and intellectual disabilities are independently associated with poor mental and general health, in children and adults.

**Design** Cross-sectional study. For Scotland's population, logistic regressions investigated odds of intellectual disabilities and autism predicting mental health conditions, and poor general health, adjusted for age and gender.

**Participants** 1 548 819 children/youth aged 0–24 years, and 3 746 584 adults aged more than 25 years, of whom 9396/1 548 819 children/youth had intellectual disabilities (0.6%), 25 063/1 548 819 children/youth had autism (1.6%); and 16 953/3 746 584 adults had intellectual disabilities (0.5%), 6649/3 746 584 adults had autism (0.2%). These figures are based on self-report.

**Main outcome measures** Self-reported general health status and mental health.

**Results** In children/youth, intellectual disabilities (OR 7.04, 95% CI 6.30 to 7.87) and autism (OR 25.08, 95% CI 23.08 to 27.32) both independently predicted mental health conditions. In adults, intellectual disabilities (OR 3.50, 95% CI 3.20 to 3.84) and autism (OR 5.30, 95% CI 4.80 to 5.85) both independently predicted mental health conditions. In children/youth, intellectual disabilities (OR 18.34, 95% CI 17.17 to 19.58) and autism (OR 8.40, 95% CI 8.02 to 8.80) both independently predicted poor general health. In adults, intellectual disabilities (OR 7.54, 95% CI 7.02 to 8.10) and autism (OR 4.46, 95% CI 4.06 to 4.89) both independently predicted poor general health.

**Conclusions** Both intellectual disabilities and autism independently predict poor health, intellectual disabilities more so for general health and autism more so for mental health. Intellectual disabilities and autism are not uncommon, and due to their associated poor health, sufficient services/supports are needed. This is not just due to coexistence of these conditions or just to having intellectual disabilities, as the population with autism is independently associated with substantial health inequalities compared with the general population, across the entire life course.

### Strengths and limitations of this study

► This large study is the first study to have reported on the extent to which autism and intellectual disabilities are independently associated with poor mental and general health, in children and adults.
► The study comprises a whole country population, with high participation rate (94%), and the conditions were systematically enquired about on everyone.
► A limitation is that conditions were self/proxy reports rather than in-depth diagnostic assessments.

## INTRODUCTION

Both intellectual disabilities and autism occur not uncommonly in children and adults, and can co-occur. Children and adults with intellectual disabilities have notably poorer mental and general health than other people.[1–4] This has also been reported for autistic children and adults,[5–9] although the quantity of research is limited, particularly with regard to adults. The extent of co-occurrence of intellectual disabilities in autistic people used to be considered to be as high as 50%–70%,[10] although more recent reports suggest that it may be lower, though still considerable, at about 20%.[9 11] This may in part relate to the broadening of criteria for the autism spectrum to include 'milder' autism and greater awareness about autism in children and young people in recent years, as it is well established that autism prevalence is higher in people with more severe intellectual disabilities and vice versa.[12] Autism is generally considered to be associated with poor mental health. However, the largest study to examine this in adults in a general community population found no difference in rates of mental ill health in adults with co-occurring autism and

intellectual disabilities, compared with age-gender-Down syndrome level of ability-matched adults with intellectual disabilities but no autism.[13] As intellectual disabilities and autism have tended to be studied separately, the relative extent to which being autistic, or having intellectual disabilities, accounts for their poor population health is not clear. This is important to understand, given the frequent co-occurrence of these conditions, and is important to understand in both child and adult populations, given the more recent change in co-occurrence due to higher frequency of diagnosis of autism.

The aim of this paper is to study the extent to which autism and intellectual disabilities are independently associated with poor mental and general health, in children and adults.

## METHOD
### Strengthening the Reporting of Observational Studies in Epidemiology guidelines
The Strengthening the Reporting of Observational Studies in Epidemiology checklist for cross-sectional studies was adhered to.

### Census process and variables
Scotland has performed a national Census every 10 years since 1841, the most recent being Scotland's Census, 2011. Information was collected on every resident in Scotland on the Census date, 27 March 2011. This included people in private households and also people in community residences (such as care homes, prisons and student halls of residence). In private households (typically family households), one person was responsible for completing the Census details for all the household's residents; for communal establishments, the manager was responsible for providing the information. It is a legal requirement in the UK to complete the Census. Failure to provide information or for providing false information attracted a fine of up to £1000. Non-responses were followed up by the Census team and help provided. These factors accounted for the high response rate; Scotland's Census 2011 achieved a 94% response rate.[14] The Census team adjusted for the 6% non-response rate using a Census Coverage Survey to estimate numbers and characteristics. The Census Coverage Survey included around 40 000 households; the records from it were matched with Census records, with all individuals deterministically matched to check for duplicates. Individuals estimated to be missing from the Census were then imputed, using a subset of characteristics from real individuals, including health information. This edit and imputation methodology was adapted from the Office for National Statistics rigorous and systematic guidelines, available at: http://webarchive.nationalarchives.gov.uk/20160108193745/http://www.ons.gov.uk/ons/guide-method/method-quality/survey-methodology-bulletin/smb-69/index.html and further details on the Census population estimates are available at: http://www.scotlandscensus.gov.

uk/documents/censusresults/release1b/rel1bmethodology.pdf

Full details of the methodology and other background information on Scotland's Census, 2011 are available at: http://www.scotlandscensus.gov.uk/supporting-information.

The Census included questions on demography, long-term conditions and on general health.

The question on long-term conditions enquired:

'Do you have any of the following conditions, which have lasted, or are expected to last, at least 12 months? Tick all that apply:
- ► Deafness or partial hearing loss.
- ► Blindness or partial sight loss.
- ► Learning disability (eg, Down's syndrome).
- ► Learning difficulty (eg, dyslexia)
- ► Developmental disorder (eg, autistic spectrum disorder or Asperger's syndrome).
- ► Physical disability.
- ► Mental health condition.
- ► Long-term illness, disease or condition.
- ► Other condition, please write in
  - – free-text space was then provided for conditions to be listed.
- ► No condition'.

The question on general health enquired:

'How is your health in general?
- ► Very good.
- ► Good.
- ► Fair.
- ► Bad.
- ► Very bad'.

The terminology used in both these questions was specifically investigated prior to implementation of data collection. The General Register Office for Scotland commissioned Ipsos MORI Scotland to undertake cognitive question testing, to determine whether the questions were answered accurately and willingly by respondents, and what changes if any might be required to improve data quality and/or the acceptability of the response options. Cognitive interviewing is a widely used approach to critically evaluate survey questionnaires.[15] It tests the way respondents understand, mentally process and respond to survey materials. It enables researchers to modify survey material to enhance clarity. Retrospective probing was deemed to be the most appropriate of the different techniques available. It involved the interviewer presenting the question, the respondent answering it, and the interviewer then probing for specific information relevant to the question or to the specific answer given (eg, What does this question mean in your own words?). This research was undertaken with 102 participants with a mix of gender and age, both with and without the health conditions and disabilities (including people with more than one of the conditions). This included people with autism, intellectual disabilities, dyslexia, dyspraxia, speech impairment, mental health conditions (both milder and more serious) and other long-term conditions. The

results found that the question on general health status functioned well and did not need amendment, as did the questions on long-term conditions, including intellectual disabilities and mental health condition, while the question on autism was redesigned to that listed above in order to more accurately capture the data specifically on autism. Additionally, the response 'no' was amended to 'no condition'. The other questions did not require any modification. Further information can be found at: http://www.scotlandscensus.gov.uk/documents/research/2011-census-health-disability-questions.pdf http://www.scotlandscensus.gov.uk/documents/legislation/changes-to-gov-statement-report.pdf

In Scotland, the term 'learning disability' is synonymous with the international term 'intellectual disabilities'.[16 17]

For 2.6% of the Census returns, information on long-term conditions was not completed. The Census team assumed the most plausible explanation was that the person had no long-term condition but did not see the 'No condition' check box at the end of the question. They, thus, recorded them to have none of the long-term conditions.

### Data analysis

First, frequency data were generated. Next, we used logistic regressions to calculate the ORs with 95% CIs of autism, intellectual disabilities, age and gender in predicting (1) having a mental health condition and (2) poor general health. We dichotomised the general health status variable to good health (very good or good health) or poor health (fair, bad, or very bad health). The gender variable was binary, the reference group was male. We conducted the analyses separately for children and young people (aged 0–24 years) and adults (aged 25+ years). This was because in Scotland's Census, 2011, the prevalence of autism is higher in the children and young people than in the adults, most likely due to widening out of the diagnostic criteria and greater awareness of autism in recent decades. Hence the adults with autism are more likely to be on the more severely affected range of the autism spectrum. For the children and young people, the reference group was aged 0–15 years (childhood), given the physiological changes and changing life experiences that occur in adolescence/transition compared with younger children, which may have a bearing on general and mental health. The adults were grouped into 10-year age bands, with the reference group being aged 25–34 years. We then conducted a second round of the regressions, including the interaction terms age x intellectual disabilities and age x autism. This was because the influence of age on mental health and general health is likely to differ in people with intellectual disabilities and possibly in people with autism to that seen in other people. All analyses were conducted with SPSS software V.22.

### Patient and public involvement

The question on intellectual disabilities and autism was included in Scotland's Census, 2011 at the behest of third sector organisations for people with intellectual disabilities and autism. This study was undertaken by the Scottish Learning Disabilities Observatory, which has a specific remit for people with intellectual disabilities and autism; its steering group includes partners from the third sector organisations. Results from this study will be disseminated for people with intellectual disabilities and autism in easy-read version via the Scottish Learning Disabilities Observatory website and newsletters.

## RESULTS

Scotland's Census, 2011, includes records on 5 295 403 people aged more than 0–75 years, of whom 1 548 819 (29.2%) were children and young people, and 3 746 584 (70.8%) were adults aged 25 years and over. Of the children and young people, 9396 (0.6%) reported having intellectual disabilities and 25 063 (1.6%) reported having autism. Of the adults aged 25 years and over, 16 953 (0.5%) reported having intellectual disabilities and 6649 (0.2%) reported having autism. Of the children and young people with intellectual disabilities, 3756/9396 (40.0%) additionally had autism, and of the adults aged 25 years and over with intellectual disabilities, 1953/16 953 (11.5%) additionally had autism. Of the children and young people with autism, 3756/25 063 (15.0%) additionally had intellectual disabilities, and of the adults aged 25 years and over with autism, 1953/6649 (29.4%) additionally had intellectual disabilities.

538/5640 (9.5%) of the children and young people with intellectual disabilities but no autism had a mental health condition, and 3383/15 000 (22.6%) of the adults with intellectual disabilities but no autism had a mental health condition. A total of 1601/21 307 (7.5%) of the children and young people with autism but no intellectual disabilities had a mental health condition, and 1314/4696 (28.0%) of adults with autism but no intellectual disabilities had a mental health condition. A total of 15 829/1 518 116 (1.0%) of the children and young people with neither condition had a mental health condition, and 208 493/3 724 935 (5.6%) of the adults with neither condition had a mental health condition.

Table 1 presents the OR (95% CI) of intellectual disabilities, autism, age and gender in predicting a mental health condition in the children and young people. It presents the results of two regressions, the second one including the interaction terms. Both intellectual disabilities (OR 7.0, 95% CI 6.3 to 7.9) and autism (OR 25.1, 95% CI 23.0 to 27.3) independently increased the odds of having a mental health condition, more so for autism. Mental health conditions were also predicted by female gender (OR 1.5, 95% CI 1.4 to 1.5) and being a young person rather than a child (OR 10.5, 95% CI 10.1 to 11.0).

In adults (table 2), a similar pattern was seen with both intellectual disabilities (OR 3.5, 95% CI 3.2 to 3.8) and autism (OR 5.3, 95% CI 4.8 to 5.9) independently predicting a mental health condition, as did female gender (OR 1.3, 95% CI 1.2 to 1.3). All age groups had

**Table 1** Predictors of mental health conditions in the whole population of children and young people

| Variable | | Regression 1 | | Regression 2 (including interaction terms) | |
|---|---|---|---|---|---|
| | | OR | 95% CI | OR | 95% CI |
| Age | 0–15 (ref) | – | | – | |
| | 16–24 | 7.65 | 7.36 to 7.95 | 10.54 | 10.06 to 11.05 |
| Gender | Male (ref) | – | | – | |
| | Female | 1.49 | 1.45 to 1.54 | 1.48 | 1.44 to 1.53 |
| Autism | No autism (ref) | – | | – | |
| | Autism | 10.21 | 9.67 to 10.78 | 25.08 | 23.02 to 27.32 |
| Intellectual disabilities | No intellectual disabilities (ref) | – | | – | |
| | Intellectual disabilities | 5.85 | 5.44 to 6.29 | 7.04 | 6.30 to 7.87 |
| Age x intellectual disabilities | 0–15 (ref) | – | | – | |
| | 16–24 | – | | 0.66 | 0.57 to 0.76 |
| Age x autism | 0–15 (ref) | – | | – | |
| | 16–24 | – | | 0.24 | 0.21 to 0.26 |
| Constant | | 0.00 | | 0.00 | |

higher odds ratios than the 25–34 years of predicting having a mental health condition, except for the oldest age group, aged 65+ years who had a lower rate.

A total of 2453/5640 (43.5%) of children and young people with intellectual disabilities but no autism, and 7834/15 000 (52.2%) of adults with intellectual disabilities

**Table 2** Predictors of mental health conditions in the whole population of adults

| Variable | | Regression 1 | | Regression 2 (including interaction terms) | |
|---|---|---|---|---|---|
| | | OR | 95% CI | OR | 95% CI |
| Age | 25–34 (ref) | – | | – | |
| | 35–44 | 1.40 | 1.38 to 1.42 | 1.40 | 1.38 to 1.42 |
| | 45–54 | 1.38 | 1.36 to 1.40 | 1.38 | 1.36 to 1.40 |
| | 55–64 | 1.08 | 1.06 to 1.09 | 1.07 | 1.05 to 1.08 |
| | 65+ | 0.92 | 0.90 to 0.93 | 0.91 | 0.90 to 0.92 |
| Gender | Male (ref) | – | | – | |
| | Female | 1.25 | 1.24 to 1.26 | 1.25 | 1.24 to 1.26 |
| Autism | No autism (ref) | – | | – | |
| | Autism (ref) | 5.29 | 5.00 to 5.59 | 5.30 | 4.80 to 5.85 |
| Intellectual disabilities | No intellectual disabilities (ref) | | | – | |
| | Intellectual disabilities | 4.42 | 4.26 to 4.59 | 3.50 | 3.20 to 3.84 |
| Age x intellectual disabilities | 25–34 (ref) | – | | – | |
| | 35–44 | – | | 0.99 | 0.87 to 1.11 |
| | 45–54 | – | | 1.24 | 1.10 to 1.39 |
| | 55–64 | – | | 1.71 | 1.51 to 1.94 |
| | 65+ | – | | 1.82 | 1.59 to 2.08 |
| Age x autism | 25–34 (ref) | – | | – | |
| | 35–44 | – | | 1.03 | 0.89 to 1.19 |
| | 45–54 | – | | 0.94 | 0.80 to 1.10 |
| | 55–64 | – | | 1.02 | 0.84 to 1.25 |
| | 65+ | – | | 1.18 | 0.97 to 1.44 |
| Constant | | 0.46 | | 0.46 | |

**Table 3** Predictors of poor general health in the whole population of children and young people

| Variable | | Regression 1 | | Regression 2 (including interaction terms) | |
|---|---|---|---|---|---|
| | | OR | 95% CI | OR | 95% CI |
| Age | 0–15 (ref) | – | | – | |
| | 16–24 | 2.14 | 2.10 to 2.18 | 2.28 | 2.24 to 2.33 |
| Gender | Male (ref) | – | | – | |
| | Female | 1.11 | 1.09 to 1.14 | 1.11 | 1.09 to 1.13 |
| Autism | No autism (ref) | – | | – | |
| | Autism | 6.70 | 6.46 to 6.95 | 8.40 | 8.02 to 8.80 |
| Intellectual disabilities | No intellectual disabilities (ref) | – | | – | |
| | Intellectual disabilities | 14.05 | 13.39 to 14.73 | 18.34 | 17.17 to 19.58 |
| Age x intellectual disabilities | 0–15 (ref) | – | | – | |
| | 16–24 | – | | 0.57 | 0.52 to 0.63 |
| Age x autism | 0–15 (ref) | – | | – | |
| | 16–24 | – | | 0.54 | 0.50 to 0.58 |
| Constant | | 0.02 | | 0.02 | |

but no autism had poor general health. A total of 3898/21 307 (18.3%) of children and young people with autism but no intellectual disabilities, and 2 134/4 696 (45.4%) of adults with autism but no intellectual disabilities had poor general health. A total of 42 713/1 518 116 (2.8%) of the children and young people with neither condition, and 880 044/3 724 935 (23.6%) of the adults with neither condition had poor general health.

Table 3 presents the OR (95% CI) of intellectual disabilities, autism, age and gender in predicting poor general health in the children and young people. It presents the results of two regressions, the second one including the interaction terms. Both intellectual disabilities (OR 18.3, 95% CI 17.2 to 19.6) and autism (OR 8.4, 95% CI 8.0 to 8.8) independently increased the odds of having poor general health, more so for intellectual disabilities. Poor general health was also predicted by female gender (OR 1.1, 955 CI 1.1 to 1.1) and being a young person rather than a child (OR 2.3, 95% CI 2.2 to 2.3.

In adults (table 4), a similar pattern was seen with both intellectual disabilities (OR 7.5, 95% CI 7.0 to 8.1 and autism OR 4.5, 95% CI 4.1 to 4.9) independently predicting poor general health, as did female gender (OR 1.1, 95% CI 1.1 to 1.1). A gradient is seen, with older age groups progressively predicting having poor general health.

## DISCUSSION
### Principal findings and interpretation
This is the largest study to date on this topic, comprising a whole country population. Our findings have demonstrated that both intellectual disabilities and autism are associated with having a mental health condition and with poor general health. This is so in both children/young people and in adults, after the overlap between these two conditions (intellectual disabilities and autism) is accounted for. For mental health conditions, this is particularly so for autism (OR 25.1, 95% CI 23.0 to 27.3 for children/young people; OR 5.3, 95% CI 4.8 to 5.9 for adults). For poor general health, this is particularly so for intellectual disabilities (OR18.3, 95% CI 17.2 to 19.6 for children/young people; OR 7.5, 95% CI 7.0 to 8.1 for adults). Previous literature on this is limited, and has not taken account of the overlap between autism and intellectual disabilities. It is of particular note that autism contributes to poor general health and especially to having a mental health condition even after taking account of the contribution of intellectual disabilities. The mental health conditions did not include transient common mental disorders, as the question referred to mental health conditions lasting or expected to last at least 12 months, that is, severe mental health conditions.

The extent of mental and general health inequality experienced by the population with intellectual disabilities and the population with autism, in comparison with the general population, is greatest in children/young people than it is for adults, though is substantial at all ages. This reflects that both mental health conditions and poor general health are much more common in adults than children and young people in the general population, while they are common at all ages in people with autism and in people with intellectual disabilities. Indeed, in people with intellectual disabilities, those with more severe intellectual disabilities have more comorbidity[18] and die at an earlier age[19] including in childhood. Hence, with increasing age, although acquiring age-related conditions, the population with intellectual disabilities has less disability-related comorbidity and perversely may be healthier than the younger population with intellectual disabilities.

**Table 4** Predictors of poor general health in the whole population of adults

| Variable | | Regression 1 | | Regression 2 (including interaction terms) | |
|---|---|---|---|---|---|
| | | OR | 95% CI | OR | 95% CI |
| Age | 25–34 (ref) | – | | – | |
| | 35–44 | 1.78 | 1.76 to 1.80 | 1.79 | 1.77 to 1.81 |
| | 45–54 | 2.86 | 2.83 to 2.89 | 2.90 | 2.87 to 2.93 |
| | 55–64 | 4.81 | 4.76 to 4.86 | 4.88 | 4.82 to 4.93 |
| | 65+ | 10.25 | 10.15 to 10.36 | 10.39 | 10.29 to 10.50 |
| Gender | Male (ref) | – | | – | |
| | Female | 1.05 | 1.05 to 1.06 | 1.05 | 1.05 to 1.06 |
| Autism | No autism (ref) | – | | – | |
| | Autism (ref) | 3.39 | 3.21 to 3.58 | 4.46 | 4.06 to 4.89 |
| Intellectual disabilities | No intellectual disabilities (ref) | | | – | |
| | Intellectual disabilities | 4.39 | 4.25 to 4.53 | 7.54 | 7.02 to 8.10 |
| Age x intellectual disabilities | 25–34 (ref) | – | | – | |
| | 35–44 | – | | 0.72 | 0.65 to 0.79 |
| | 45–54 | – | | 0.60 | 0.54 to 0.65 |
| | 55–64 | – | | 0.45 | 0.40 to 0.50 |
| | 65+ | – | | 0.24 | 0.21 to 0.26 |
| Age x autism | 25–34 (ref) | – | | – | |
| | 35–44 | – | | 0.83 | 0.72 to 0.96 |
| | 45–54 | – | | 0.59 | 0.50 to 0.68 |
| | 55–64 | – | | 0.49 | 0.41 to 0.59 |
| | 65+ | – | | 0.44 | 0.36 to 0.53 |
| Constant | | 0.08 | | 0.08 | |

The aetiology of mental and general ill health in people with intellectual disabilities or people with autism includes genetic predisposition[18]; indeed the term, Early Symptomatic Syndromes Eliciting Neurodevelopmental Clinical Examination, has been coined for the association of problems in one or more of 10 health domains in young children.[19] It is clear though that aetiology is multifactorial, and social and environmental factors such as life events, which occur more commonly in people with intellectual disabilities, have been shown to precede onset of mental health conditions in adults with intellectual disabilities.[20] Therefore, underpinning mechanisms appear to include the interplay between genes, environment and lifestyle[21] including differences in health related behaviours, such as diet and exercise,[22] and inequalities in access to services.[23] This is important, as understanding these factors provides pathways to the development of interventions to improve health. Examples tailored to these populations, in addition to drug treatments, include interventions developed to address lifestyle,[24] general health[25] and psychological interventions for mental health conditions.[26]

### Comparison with previous literature
No previous studies have been identified which investigated the extent to which autism and intellectual disabilities are independently associated with poor mental and general health, in children and adults. We believe that this is, therefore, the first study to do so and subsequently we cannot compare these results.

### Strengths and limitations
The main strengths of the study are the 94% whole population response, rather than biassed sampling; the large population size of 5.3 million; that the conditions (intellectual disabilities, autism, mental health condition and general health) were systematically enquired about for each person; and that the phrasing of the questions underwent cognitive question testing in advance of the Census to ensure they captured the intended meaning. Consequently, we believe that these results are generalisable to other high-income countries.

Limitations include the use of the term of 'developmental disorders' in the Census. However, the Census form prompted responses only for autistic spectrum disorder or Asperger's syndrome. Furthermore, the developmental disorders category was distinguished from intellectual disabilities, learning difficulties and mental health conditions, which are important distinctions. Hence, we consider that respondents will have replied accordingly, that is, responded regarding autism.

However, we have no means to check this. In addition, conditions were self/proxy reports rather than in-depth diagnostic assessments (which would not be possible on such a large scale). Respondents reported whether or not each person was known to have autism and/or intellectual disabilities, rather than each person having an assessment, so some reporting error is possible. However, intellectual disabilities and autism are conditions that are typically diagnosed during infant/primary school age, if not before. In Scotland these diagnoses attract additional educational support, which is to the child's advantage; once diagnosed these are lifetime diagnoses. Consequently, there may be an undercount in the early years of childhood, whereas reporting of these conditions should be accurate in later childhood, youth and in adults, within the diagnostic criteria prevailing at the time of diagnosis. The proportion of people in the population reported to have autism was lower after age 25. This reflects the broadening of diagnostic criteria and greater awareness of autism in recent years; hence, the older people with autism might have more severe autism than the children/youth reported to have autism. The children/youth with autism are likely to include some who function well. We do not know the extent to which reporting of mental health conditions and general health status would reflect that found in in-depth diagnostic assessments, although subjective general health status is commonly used in population studies, and it is well established as an extremely valid measure of health. There is a strongly predictive linear gradient across subjective health status and subsequent number of medical appointments, hospital admissions and mortality.[27–29] We do not know the proportion who self-reported or for whom the report was by another household reference person (eg, parent). However, the latter is likely to be more common for the people with intellectual disabilities in view of their intellectual disabilities, and for the children and young people. Six per cent of Census entries were imputed. The Census team assumed the 2.6% who did not provide information on long-term conditions did not have any of them, but we are unable to confirm the accuracy of this assumption.

Future research investigating narrower age bands of children/youth may be useful, and next steps must importantly include study of the aetiology of poor health in these populations, to inform the development of further effective interventions.

## IMPLICATIONS

Intellectual disabilities and autism are not uncommon, and due to their associated poor mental and general health, services and supports need to be available in sufficient quantity and quality. Our findings demonstrate that this is not just related to the coexistence of these conditions, or just to having intellectual disabilities, as the population with autism is also independently associated with substantial health inequalities compared with the general population, across the entire life course.

**Acknowledgements** We acknowledge funding from the UK Medical Research Council (grant number: MC_PC_1717).

**Contributors** DK analysed the data, jointly interpreted it and wrote the first draft of the manuscript. ER jointly interpreted the data and contributed to the manuscript. KD jointly interpreted the data and contributed to the manuscript. LAH-M jointly interpreted the data and contributed to the manuscript. CM jointly conceived the project, interpreted the data and contributed to the manuscript. AH jointly interpreted the data and contributed to the manuscript. S-AC jointly conceived the project, interpreted the data and contributed to the manuscript. All authors approved the final version of the manuscript. S-AC is the study guarantor.

**Funding** This study was funded by the Medical Research Council (grant reference MC_PC_17217) and the Scottish Government via the Scottish Learning Disabilities Observatory.

**Disclaimer** The funders had no role in the study design, collection, analyses and interpretation of data, in writing the report, nor in the decision to submit the article for publication.

**Competing interests** None declared.

**Patient consent for publication** Not required.

**Ethics approval** Approval was gained from the Scottish Government for secondary analysis of the Scotland Census, 2011 data. Access to a subset of data was provided.

**Provenance and peer review** Not commissioned; externally peer reviewed.

**Data availability statement** No data are available.

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
