## [Reviewer comments · BMJ Open]

ARTICLE DETAILS

TITLE (PROVISIONAL)	The relative influence of intellectual disabilities and autism on mental and general health in Scotland. A cross-sectional study of a whole country of 5.3 million children and adults
AUTHORS	Kinnear, Deborah; Rydzewska, Ewelina; Dunn, Kirsty; Hughes-McCormack, Laura; Melville, Craig; Henderson, Angela; Cooper, Sally-Ann

VERSION 1 – REVIEW

REVIEWER	Hanne Marit Bjoergaas Dept. of Child- and Adolescent Psychiatry Stavanger University Hospital Norway
REVIEW RETURNED	30-Jan-2019

GENERAL COMMENTS	This paper addresses an important issue; health conditions (both mental and general health) in vulnerable groups in the society, and should be published. There are a couple of concerns which should be addressed. Firstly, ethical issues related to consent to participate in the study is slightly unclear to me, and I would like the following question to be elaborated: Is it realistically an option not to partake in the study, and how would that option be granted when one person was responsible for completing the census details for all the household's residents? The study limitations are well described, and also the possible reasons for a very significant difference in the rate of autism spectrum disorders in the child and adolescent population compared to the adult population. Perhaps the wide spectre of autism spectrum disorders, including otherwise well-functioning young people with autism spectrum disorders should be addressed in this context? Further, the limitations of self-report should perhaps also be elaborated more in the discussion. Lastly, I would like to point to a minor concern; shorter and more concise sentences would be more reader-friendly. To conclude: the paper addresses an issue that is of importance when planning both mental health care and general health care for vulnerable groups in our society.
---

REVIEWER	Jon Quach The University of Melbourne
REVIEW RETURNED	01-Mar-2019

GENERAL COMMENTS	This is an interesting paper looking at the relationship between intellectual disabilities and autism with mental and general health. A few areas which require clarification are described below  - In the introduction, what is the justification for looking at both adults and children in the same study? Is it assumed the relationship will be the same or the mechanism of influence? - How was Autism and Intellectual disabilities specifically defined or measured? From the presented measure, it states "Developmental disorder (for example, ASD). This doesn't seem to capture Autism specifically. - A reference is required to support the statement "In Scotland, the term 'learning disability' is synonymous with the international term 'intellectual disabilities'. Is this supported by public understanding of the terms? - How was mental health measured, this is also not clear from the methods section. - As there is large overlap between individuals with ID and ASD, it is suggested that analyses could be repeated for ASD only, ID only and combined ASD/ID. This will better represent the unique contribution based on classification. In the results, it would be clearer to present the percentages in absolute terms, instead of 9.5% of children with ID had no ASD. Does it equate to 9.5% of 0.6%?  - For the statistical analyses, it is developmental inappropriate to compare 0-15 year olds to 16 - 24 year olds. What is the justification for not using smaller year bands given the rapid changes in health and development in younger children, as well as age of diagnosis. This would effect the interaction analyses for Regression 2 - For the discussion, can the authors discuss the mechanisms by which ASD and ID influence mental and general health outcomes? Does the research support age differences? Although there isn't research which has directly examined the exact relationship studied here, examining previous research could provide guidance on the mechanisms in which clinicians could focus on when supporting these individuals - Have any intervention studies been conducted which have shown addressing that mental or general health needs of individuals with ASD or ID leads to better outcomes? - The conclusion is missing
---

REVIEWER	Jason Schnittker University of Pennsylvania
REVIEW RETURNED	14-Mar-2019

GENERAL COMMENTS	This study uses census data from Scotland to estimate relationships between two disorders, autism and intellectual disability, and both physical and mental health problems, defined generally. The large dataset is certainly an asset and the topic is interesting and important--the study is well-worth undertaking and I enjoyed reading it. But the authors repeated reference to how common the two disorders are is a little self-defeating, as is the refrain that the two disorders contribute substantially to health inequality. In the full sample no more than one-half of one percent has an intellectual disability or autism. And we learn from the descriptive discussion
---

	that the vast majority of people with either conditions do not report a mental health condition or poor self-rated health. The contributions of these two conditions to poor population health writ large are almost certainly very small. At least there is no empirical basis to infer that autism is associated with “substantial health inequalities,” as the authors do in the implications section. Autism is correlated with poor health, as the authors show, but that is not the same thing as entertaining autism's contribution to poor population health. At the same time I'm not sure the authors make a clear contribution relative to what has been published. There is a good deal of comorbidity between autism and other psychiatric disorders, as is well known. In this study we learn that there is an association between autism and some other unspecified "mental health condition"(s), but we don't know what that condition(s) is, nor do we understand the process that led to the comorbidity. The questions in the census refer to diagnosed prevalence, or at least "mental conditions" the respondent recognizes as such, so one explanation is simply that the person who diagnosed the respondent with autism also diagnosed them with another disorder, like depression. By the same token, there are relationships between autism and intellectual disability and poor self-reported health, as the authors report, but this does not have a plain interpretation. When reporting self-rated health in surveys respondents consider psychiatric disorders, so part of the association the authors find might simply be that those with autism report poor health precisely because and only because they have been diagnosed with autism. Of course, there are other possibilities, including that intellectual disabilities lead to poorer environmental conditions, like low SES, that in turn lead to an increase in poor health. But the models include only four characteristics as independent variables so we don't learn much about the reasons for the comorbidity. In the same vein, it's useful to learn that intellectual disabilities and autism contribute independently to poor health, but we never actually learn about the nature of the overlap between the two. We never see two-by-two tables. Do they contribute independently because they are not, in fact, highly correlated? We also don't really know whether autism specifically is what the person was diagnosed with. The census question refers to a "developmental disorder," for which Autistic Spectrum Disorder is only given as an example. How confident should we be that autistic people are placing themselves in this category rather than “learning disability” or “learning difficulty” or that the "development disorder" respondents report is not something else? The cognitive testing the authors review does not address this issue directly. That testing mostly pertains, it seems, to whether people with assorted conditions are willing to positively respond to health questions as those questions are asked in the census.
--	---

VERSION 1 – AUTHOR RESPONSE

Reviewer: 1

This paper addresses an important issue; health conditions (both mental and general health) in vulnerable groups in the society, and should be published. There are a couple of concerns which should be addressed.

Firstly, ethical issues related to consent to participate in the study is slightly unclear to me, and I would like the following question to be elaborated:

Is it realistically an option not to partake in the study, and how would that option be granted when one person was responsible for completing the census details for all the household's residents?

***Response: It is a legal requirement in the UK to complete the Census, and in Scotland 94% did so. We were granted approval to conduct secondary analysis of the Census data, in the public interest. This information is provided in the methods section (page 3) "It is a legal requirement in the UK to complete the Census. Failure to provide information, or for providing false information attracted a fine of up to £1,000. Non-responses were followed up by the Census team and help provided. These factors accounted for the high response rate; Scotland's Census 2011 achieved a 94% response rate."

The study limitations are well described, and also the possible reasons for a very significant difference in the rate of autism spectrum disorders in the child and adolescent population compared to the adult population. Perhaps the wide spectre of autism spectrum disorders, including otherwise well-functioning young people with autism spectrum disorders should be addressed in this context?

***Response: We have expanded this section of the discussion on study limitations, so that it is more prominent (page 8), as follows:

"The proportion of people in the population reported to have autism was lower after age 25. This reflects the broadening of diagnostic criteria and greater awareness of autism in recent years; hence the older people with autism might have more severe autism than the children/youth reported to have autism. The children/youth with autism are likely to include some who function well."

Further, the limitations of self-report should perhaps also be elaborated more in the discussion.

***Response: We have expanded this section of the discussion on study limitations, so that it is considered in more detail (page 8), as follows:

"In addition, conditions were self/proxy reports rather than in-depth diagnostic assessments (which would not be possible on such a large scale). Respondents reported whether or not each person was known to have autism and/or intellectual disabilities, rather than each person having an assessment, so some reporting error is possible. However, intellectual disabilities and autism are conditions that are typically diagnosed during infant/primary school age, if not before. In Scotland these diagnoses attract additional educational support which is to the child's advantage; once diagnosed these are lifetime diagnoses."

Lastly, I would like to point to a minor concern; shorter and more concise sentences would be more reader-friendly.

***Response: We have shortened sentences throughout to try and make more reader-friendly.

To conclude: the paper addresses an issue that is of importance when planning both mental health care and general health care for vulnerable groups in our society.

Reviewer: 2

This is an interesting paper looking at the relationship between intellectual disabilities and autism with mental and general health. A few areas which require clarification are described below

- In the introduction, what is the justification for looking at both adults and children in the same study? Is it assumed the relationship will be the same or the mechanism of influence?

***Response: We wanted to determine the relative extent that autism and intellectual disabilities are independently associated with poor mental and general health. We found no previous literature to indicate whether findings would be similar or different in children and adults, and are aware of the change in diagnostic criteria and awareness of the autism spectrum, that influenced the rates of autism that are present in the Census for children/youth, versus adults over 25 years. We have amended the last sentence of the first paragraph of the introduction to address this (page 2), as follows:

“As intellectual disabilities and autism have tended to be studied separately, the relative extent to which being autistic, or having intellectual disabilities, accounts for their poor population health is not clear. This is important to understand, given the frequent co-occurrence of these conditions, and is important to understand in both child and adult populations, given the more recent change in co-occurrence due to higher frequency of diagnosis of autism.”

- How was Autism and Intellectual disabilities specifically defined or measured? From the presented measure, it states "Developmental disorder (for example, ASD). This doesn't seem to capture Autism specifically.

***Response: Both were measured by self/proxy report. The wording of the questions was specifically tested in advance using cognitive question testing, and in the case of autism, this methodological development resulted in the question on autism being redesigned to the one used, to more accurately capture data specifically on autism. We provide more details on this on page 4. We also have now expanded the limitations section (page 8) as outlined below:

“Limitations include the use of the term of ‘developmental disorders’ in the Census. However, the Census form prompted responses only for autistic spectrum disorder or Asperger’s syndrome. Furthermore, the developmental disorders category was distinguished from intellectual disabilities, learning difficulties, and mental health conditions, which are important distinctions. Hence, we consider that respondents will have replied accordingly, i.e. responded regarding autism. However, we have no means to check this. In addition, conditions were self/proxy reports rather than in-depth diagnostic assessments (which would not be possible on such a large scale). Respondents reported whether or not each person was known to have autism and/or intellectual disabilities, rather than each person having an assessment, so some reporting error is possible. However, intellectual disabilities and autism are conditions that are typically diagnosed during infant/primary school age, if not before. In Scotland these diagnoses attract additional educational support which is to the child’s advantage; once diagnosed these are lifetime diagnoses.”

- A reference is required to support the statement "In Scotland, the term 'learning disability' is synonymous with the international term 'intellectual disabilities'. Is this supported by public understanding of the terms?

***Response: We have now added a reference to a Scottish learning disabilities strategy ‘Keys to Life’ (Scottish Government, 2013) as well as a recent report on health assessment of people with learning disabilities in Scotland produced by NHS Health Scotland (Truesdale and Brown, 2017) (page 4). Both these reports provide explanation of the term ‘learning disabilities’, which is typically used in Scotland and is synonymous with the term ‘intellectual disabilities’.

- How was mental health measured, this is also not clear from the methods section.

***Response: Mental health was measured through self/proxy-report which we have now made clear in the abstract (page 1)

- As there is large overlap between individuals with ID and ASD, it is suggested that analyses could be repeated for ASD only, ID only and combined ASD/ID. This will better represent the unique contribution based on classification.

***Response: The paper already includes 4 tables, and we feel would be cumbersome with the addition of further data. We have previously reported Census data separately on people with autism, and people with intellectual disabilities, which we reference in the paper (references number 3, 4, 7 and 9); it is because we found the overlap between intellectual disabilities and autism, that we designed this current study to investigate the relative contributions of the two conditions on the extent of poor health outcomes.

In the results, it would be clearer to present the percentages in absolute terms, instead of 9.5% of children with ID had no ASD. Does it equate to 9.5% of 0.6%?

***Response: This is a misunderstanding, and we should have presented the data more clearly, and so have rewritten as follows:

“538/5,640 (9.5%) of the children and young people with intellectual disabilities but no autism had a mental health condition, and 3,383/15,000 (22.6%) of the adults with intellectual disabilities but no autism had a mental health condition. 1,601/21,307 (7.5%) of the children and young people with autism but no intellectual disabilities had a mental health condition, and 1,314/4,696 (28.0%) of adults with autism but no intellectual disabilities had a mental health condition. 15,829/1,518,116 (1.0%) of the children and young people with neither condition had a mental health condition, and 208,493/3,724,935 (5.6%) of the adults with neither condition had a mental health condition.”

- For the statistical analyses, it is developmental inappropriate to compare 0-15 year olds to 16 - 24 year olds. What is the justification for not using smaller year bands given the rapid changes in health and development in younger children, as well as age of diagnosis. This would affect the interaction analyses for Regression 2

***Response: This was our a priori statistical analysis plan, given the physiological changes and changing life experiences that occur in adolescence/transition compared with children, which may have a bearing on general and mental health, and which we did find to actually be the case in the regressions (tables 1 and 3). This was agreed with the data controller in advance, and is in keeping with the standard age bands that the Census team use for their statistical outputs. Any further sub-analyses would require a fresh set of approvals (and the associated delays), as they would be essentially be a new project. We have added a better explanation of our rationale in the methods (page 4), and in the discussion (page 8) as follows:

“For the children and young people, the reference group was aged 0-15 years (childhood), given the physiological changes and changing life experiences that occur in adolescence/transition compared with younger children, which may have a bearing on general and mental health.”

“Future research investigating narrower age bands of children/youth may be useful.

- For the discussion, can the authors discuss the mechanisms by which ASD and ID influence mental and general health outcomes? Does the research support age differences? Although there isn't research which has directly examined the exact relationship studied here, examining previous research could provide guidance on the mechanisms in which clinicians could focus on when supporting these individuals

***Response: We agree, although we have not been able to find research on age differences. We have inserted a new paragraph in the discussion (page 7), as follows:

“The aetiology of mental and general ill-health in people with intellectual disabilities or people with autism includes genetic predisposition¹⁸; indeed the term ESSENCE (Early Symptomatic Syndromes Eliciting Neurodevelopmental Clinical Examination) has been coined for the association of problems in one or more of ten health domains in young children.¹⁹ It is clear though that aetiology is multifactorial, and social and environmental factors such as life events, which occur more commonly in people with intellectual disabilities, have been shown to precede onset of mental health conditions in adults with intellectual disabilities.²⁰ Therefore, underpinning mechanisms appear to include the interplay between genes, environment, and lifestyle,²¹ including differences in health related behaviours, such as diet and exercise,²² and inequalities in access to services.²³ This is important, as understanding these factors provides pathways to the development of interventions to improve health. Examples tailored to these populations, in addition to drug treatments, include interventions developed to address lifestyle,²⁴ general health,²⁵ and psychological interventions for mental health conditions.²⁶

- Have any intervention studies been conducted which have shown addressing that mental or general health needs of individuals with ASD or ID leads to better outcomes?

***Response: Yes, and we now have discussed this at the end of the new paragraph in the discussion, above.

- The conclusion is missing

***Response: The manuscript format does not have a section on 'Conclusions'. However, we have incorporated our conclusions in the discussion section and strengths and limitations.

Reviewer: 3

This study uses census data from Scotland to estimate relationships between two disorders, autism and intellectual disability, and both physical and mental health problems, defined generally. The large dataset is certainly an asset and the topic is interesting and important--the study is well-worth undertaking and I enjoyed reading it.

-But the authors repeated reference to how common the two disorders are is a little self-defeating, as is the refrain that the two disorders contribute substantially to health inequality. In the full sample no more than one-half of one percent has an intellectual disability or autism.

***Response: Whilst being minority populations, intellectual disabilities and autism occur more commonly than many other conditions, and we refer to them in the paper as "not uncommon" to avoid over-statement. Where we have used the term health inequalities, we mean this to refer to inequalities experienced by the population with intellectual disabilities and the population with autism, compared with the general population. We recognise there are many other contributors to inequalities within the general population, and that the total contribution to inequalities in the whole population due to intellectual disabilities or autism is small, and do not wish to give the impression that we are claiming otherwise. Given this misunderstanding, we have carefully scrutinised the language we have used throughout so that this is clearer. We have removed the term "health inequalities" from the key words, and rephrased in the discussion in the first two paragraphs and in the implications section, and in the conclusion of the abstract.

- And we learn from the descriptive discussion that the vast majority of people with either conditions do not report a mental health condition or poor self-rated health. The contributions of these two conditions to poor population health writ large are almost certainly very small. At least there is no empirical basis to infer that autism is associated with "substantial health inequalities," as the authors do in the implications section. Autism is correlated with poor health, as the authors show, but that is not the same thing as entertaining autism's contribution to poor population health.

***Response: As above, we are certainly not trying to infer that intellectual disabilities and autism accounts for a large chunk of the general population's inequalities, as it does not, and we have reviewed the language we use throughout the paper. Yes, autism is correlated with poor health: we consider the extent of this to be substantial inequalities compared with the general population, given the OR=25.08 (children) and 5.30 (adults) for mental health conditions, and 8.40 (children) and 4.46 (adults) for poor general health.

At the same time I'm not sure the authors make a clear contribution relative to what has been published. There is a good deal of comorbidity between autism and other psychiatric disorders, as is well known. In this study we learn that there is an association between autism and some other unspecified "mental health condition"(s), but we don't know what that condition(s) is, nor do we understand the process that led to the comorbidity. The questions in the census refer to diagnosed prevalence, or at least "mental conditions" the respondent recognizes as such, so one explanation is simply that the person who diagnosed the respondent with autism also diagnosed them with another disorder, like depression.

***Response: Yes, there have been previous investigations on comorbidity of mental health conditions with autism. What our study adds is a quantification on the extent to which autism and intellectual disabilities independently contribute to the co-morbidity, which is important given the degree of overlap that exists between these two conditions. We are not aware of this having been previously studied. Additionally, our study is considerably larger in scale than other published studies on autism or on intellectual disabilities, due to the unique nature of the Census in Scotland, so allows investigation of an entire country population and avoids potential bias being introduced by sampling. By the same token, there are relationships between autism and intellectual disability and poor self-reported health, as the authors report, but this does not have a plain interpretation. When reporting self-rated health in surveys respondents consider psychiatric disorders, so part of the association the authors find might simply be that those with autism report poor health precisely because and only

because they have been diagnosed with autism. Of course, there are other possibilities, including that intellectual disabilities lead to poorer environmental conditions, like low SES, that in turn lead to an increase in poor health. But the models include only four characteristics as independent variables so we don't learn much about the reasons for the comorbidity.

***Response: The responses to the questions were self-reported so inevitably they are subjective, which we acknowledge in the limitations section of the discussion. However, how these questions were completed by people with the conditions was actually tested as part of the methodological development of the Census, specifically on these questions. We have therefore expanded the detail we provided on this cognitive question testing on these questions, so that it now prominently states that this covered ratings of general health, mental health, intellectual disabilities and autism (page 4), as follows:

".....The results found that the question on general health status functioned well and did not need amendment, as did the question on long-term conditions, including intellectual disabilities and mental health condition, whilst the question on autism was redesigned to that listed above in order to more accurately capture the data specifically on autism."

We did not design the study to investigate the causes of poor health in these populations; as a starting point we designed the study to investigate the extent of poor health that intellectual disabilities and autism were individually associated with. We included age and gender within the regressions given that both age-structure (younger), and gender-structure (more males) differ in the population with intellectual disabilities and the population with autism, compared with the general population. We fully agree that understanding aetiology is also important, and we have added an additional paragraph in the discussion on page 7, which highlights that it is important to understand aetiology in order to be able to develop effective interventions. Additionally, we have added at the end of the introduction (page 8) that research on aetiology is important and needed, as follows:

"next steps must importantly include study of the aetiology of poor health in these populations, to inform the development of further effective interventions."

In the same vein, it's useful to learn that intellectual disabilities and autism contribute independently to poor health, but we never actually learn about the nature of the overlap between the two. We never see two-by-two tables. Do they contribute independently because they are not, in fact, highly correlated?

***Response: There is quite a degree of overlap, as expected, and the data also confirms the shift in diagnosis of the autism spectrum between the children and adults. We have added the figures into the results (page 5) as follows:

"Of the children and young people with intellectual disabilities, 3,756/9,396 (40.0%) additionally had autism, and of the adults aged 25 years and over with intellectual disabilities, 1,953/16,953 (11.5%) additionally had autism. Of the children and young people with autism, 3,756/25,063 (15.0%) additionally had intellectual disabilities, and of the adults aged 25 years and over with autism, 1,953/6,649 (29.4%) additionally had intellectual disabilities."

-We also don't really know whether autism specifically is what the person was diagnosed with. The census question refers to a "developmental disorder," for which Autistic Spectrum Disorder is only given as an example. How confident should we be that autistic people are placing themselves in this category rather than "learning disability" or "learning difficulty" or that the "development disorder" respondents report is not something else? The cognitive testing the authors review does not address this issue directly. That testing mostly pertains, it seems, to whether people with assorted conditions are willing to positively respond to health questions as those questions are asked in the census.

***Response: The wording on the Census was:

- Learning disability (for example, Down's Syndrome)
- Learning difficulty (for example, dyslexia)
-
- Developmental disorder (for example, Autistic Spectrum Disorder or Asperger's Syndrome)

We think it is a strength that all three conditions were included and that the examples were given to distinguish them. This was also tested in the cognitive question testing with people with these disorders. The cognitive testing was to check that the questions were answered accurately, as well as willingly, and what changes might be required to improve data quality and/or the acceptability of the question. We have expanded the description of this, so it is clearer (page 4), as follows: “

“Cognitive interviewing is a widely used approach to critically evaluate survey questionnaires.¹⁵ It tests the way respondents understand, mentally process, and respond to survey materials. It enables researchers to modify survey material to enhance clarity. Retrospective probing was deemed to be the most appropriate of the different techniques available. It involved the interviewer presenting the question, the respondent answering it, and the interviewer then probing for specific information relevant to the question or to the specific answer given (e.g. What does this question mean in your own words?).”

We also acknowledge the issue in the limitation section of the discussion (pages 7/8):

‘Limitations include the use of the term of ‘developmental disorders’ in the Census. However, the Census form prompted responses only for autistic spectrum disorder or Asperger’s syndrome. Furthermore, the developmental disorders category was distinguished from intellectual disabilities, learning difficulties, and mental health conditions, which are important distinctions. Hence, we consider that respondents will have replied accordingly, i.e. responded regarding autism. However, we have no means to check this..’